# Understanding the Interaction between Nanomaterials Originated from High-Temperature Processed Starch/Myristic Acid and Human Monocyte Cells

**DOI:** 10.3390/foods13040554

**Published:** 2024-02-12

**Authors:** Vaiyapuri Subbarayan Periasamy, Jegan Athinarayanan, Ali A. Alshatwi

**Affiliations:** Department of Food Science and Nutrition, College of Food Science and Agriculture, King Saud University, P.O. Box 2460, Riyadh 11451, Saudi Arabia; psubbarayan@ksu.edu.sa (V.S.P.); jegan@ksu.edu.sa (J.A.)

**Keywords:** carbohydrates, fatty acid, toxicology, nanomaterials, immune, gene expression

## Abstract

High-temperature cooking approaches trigger many metabolically undesirable molecule formations, which pose health risks. As a result, nanomaterial formation has been observed while cooking and reported recently. At high temperatures, starch and myristic acid interact and lead to the creation of nanomaterials (cMS-NMs). We used a non-polar solvent chloroform to separate the nanomaterials using a liquid–liquid extraction technique. The physico-chemical characterization was carried out using dynamic light scattering (DLS), transmission electron microscopy (TEM), thermogravimetric analysis (TGA), and Fourier-transform infrared spectroscopy (FTIR). To determine the biological impact of these nanomaterials using different in vitro assays, including a cell viability assay, microscopic staining, and gene expression analysis, we adopted the THP-1 cell line as an in vitro monocyte model in our study. The TEM images revealed that fabricated cMS nanomaterials are smaller than 100 nm in diameter. There were significant concerns found in the cytotoxicity assay and gene expression analysis. At concentrations of 100–250 µg/mL, the cMS-NMs caused up to 95% cell death. We found both necrosis and apoptosis in cMS-NMs treated THP-1 cells. In cMS-NMs-treated THP-1 cells, we found decreased expression levels in IL1B and NFKB1A genes and significant upregulation in MIF genes, suggesting a negative immune response. These findings strongly suggest that cMS-NMs originated from high-temperature food processing can cause adverse effects on biological systems. Therefore, charred materials in processed foods should be avoided in order to minimize the risk of health complications.

## 1. Introduction

Food processing at high temperatures, such as baking, frying, and roasting, not only enhances taste and texture but also maintains food’s safety and extends its shelf life. Through this process, food undergoes significant chemical changes that result in distinct flavors and textures due to Maillard reactions, caramelization, and other complex chemical reactions [1,2]. As a result of high-temperature processing, unintended metabolically unwanted chemicals may be formed. Chemical byproducts include heterocyclic amines (HCAs), polycyclic aromatic hydrocarbons (PAHs), advanced glycation end products (AGEs), and acrylamide [3,4,5]. Heterocyclic amines (HCAs) are a group of metabolically unwanted chemicals that can form when certain foods, including meat, poultry, fish, and eggs, are cooked at high temperatures. Common cooking methods like barbecuing, frying, and grilling involve high heat, which can trigger chemical reactions leading to the formation of HCAs [6,7]. During high-temperature cooking, carbonaceous PAHs can form when meat is grilled, fried, or barbecued. Fat and other food ingredients (proteins, carbohydrates, etc.) coming into contact with the heat source produce flames and smoke, carrying these carbonaceous PAHs that can attach to the meat’s surface. These PAHs are of concern as they are mutagenic and can alter DNA, potentially increasing the risk of cancer [8,9]. When food is cooked at high temperatures using methods such as grilling, frying, or baking, AGEs can form. Several studies strongly suggest that consuming high levels of AGEs may have negative effects on our health. AGEs have been associated with increased oxidative stress, inflammation, and the development of chronic diseases like diabetes and cardiovascular disease [10].

Apart from causing the formation of small molecules, food that is cooked, fried, baked, or roasted at elevated temperatures triggers the formation of nanomaterials such as carbon-based nanomaterials [11,12,13,14,15]. The unique features of carbon nanoparticles—the different surface chemistry, polarity, shape, and size of these metabolically unwanted nanomaterials—raise concerns about their impact on human health. When compared with bulk food particles, nanomaterials have a greater surface area, which allows them to interact with biological systems easily [16]. It is important to note that the effects of carbon nanomaterials on human cells can vary depending on factors like the type of nanomaterial, its size, shape, and surface properties. Several studies indicate that certain carbon nanomaterials can be harmful to cells, causing cell death, inflammation, or disruptions in cellular processes [17].

At elevated temperatures, myristic acid and starch undergo pyrolysis or partial combustion, which is a chemical decomposition reaction caused by heat and results in the breakdown of the molecular structure of the molecules. As a result of this process, a number of reactive intermediates and byproducts are generated [18]. When myristic acid and starch molecules are broken down, carbonaceous fragments are released. As these fragments undergo further chemical changes, such as cyclization, aromatization, and polymerization, they undergo even more chemical transformations [18,19]. A result of this process is that small carbon-rich clusters form, agglomerate, and coalesce, resulting in the formation of carbon-based nanoparticles. Eventually, these carbon-rich fragments are rearranged by thermal decomposition, which leads to the formation of nanoparticles through the rearrangement of these fragments and their subsequent aggregation by covalent bond [20,21]. In particular, these fragments reassemble into nanomaterials composed of carbon, exhibiting specific properties like size, surface property, and thermal stability, which are essential for understanding their behavior and potential impact on biology [22].

Surface features may facilitate interactions with biological components, such as cell membranes, nucleic acids, and proteins, resulting in changes to cellular function, signaling, and stability [22,23]. Due to nanomaterials’ unique features and small size, they can potentially interact with breach cellular membranes [24]. When this interaction occurs, the integrity of the cell membrane can be compromised, affecting its permeability and potentially allowing the entry of foreign materials into the cell or altering its function. Upon entering a cell, these nanomaterials may interact with the cytoplasmic components of the cell. There is a possibility that they may interfere with cellular organelles or molecular pathways, disrupting normal cellular functions or signaling pathways [23,25]. The viability of cells may be affected by these interactions. Carbon nanoparticles can induce cell death mechanisms depending on their concentration and exposure time. As a result of exposure to nanomaterials, immune responses may be induced within the cells. It is possible that exposure to these foreign particles will trigger immune reactions, resulting in inflammation or the activation of immune cells in the body. Due to the immune response, cytokines and other immune factors may be released [26,27,28]. Several studies suggest that carbon nanoparticles can trigger an immune response because the body recognizes them as foreign materials. This response involves the activation of immune cells like macrophages, which work to engulf and remove the nanoparticles. However, the specific nature and extent of this response depend on factors such as the size, shape, and surface chemistry of the particles. Eventually, it leads to chronic inflammation and is associated with various health issues, including cardiovascular diseases, respiratory disorders, and autoimmune conditions. Myristic acid and starch-derived carbon nanomaterials can be perceived as foreign substances by the immune system, leading to a negative immune response. This response is likely due to the distinct surface characteristics and size of these nanomaterials. The use of in vitro models provides a convenient method for studying the immunotoxicity of carbon nanomaterials. Specifically, human monocytes play a vital role in engulfing foreign substances like carbon nanomaterials and interacting with other immune cells.

Food-based nanomaterials’ behavior within human cells, their toxicological impact on cellular structures, and their effects on immune responses are not fully understood, including the possibility of interference with cellular functions and long-term consequences. Monocyte cells respond to foreign substances by modulating several inflammatory and pro-inflammatory key genes’ expression levels. As a key component of the immune response, this cytokine influences the recruitment of immune cells and the release of other inflammatory signaling molecules, which is a crucial part of initiating and amplifying the inflammatory cascade in response to potentially harmful nanomaterials [29,30,31,32,33]. Essential markers, such as IL1B, MIF, BAK, and NFKB1A genes, are crucial for regulating immune responses, inducing macrophages, and controlling cell death. These markers are important in understanding how carbon nanomaterials can impact cellular processes, potentially leading to impairments. Thus, in this study, we used myristic acid and starch as fundamental ingredients and heated them at 240 °C for a period of time. The myristic-acid- and starch-derived nanomaterials’ (cMS-NMs) potential interactions with immune cells were studied using THP-1 as an in vitro monocyte model. Additionally, we examined the cMS-NMs’ effects on cellular structure, function, viability, and molecular changes.

## 2. Materials and Methods

### 2.1. Materials

A THP-1 cell line was acquired from the American Type Culture Collection (ATCC, Manassas, VA, USA). The chemicals, including dimethyl sulfoxide (DMSO), 5-(4,5-dimethylthiazol-2-yl)-2,5-diphenyl tetrazolium bromide (MTT), and ethidium bromide (EB), were purchased from Sigma-Aldrich (St. Louis, MO, USA) and the starch and myristic acid from Nice Chemicals (Mumbai, India). Supplies of RPMI medium and fetal bovine serum (FBS) were obtained from Invitrogen (Carlsbad, CA, USA). We purchased the Fastlane Cell cDNA Kit, QuantiTect Primer Assay, and QuantiFast SYBR Green PCR Kit from Qiagen (Hilden, Germany).

### 2.2. Fabrication of Nanomaterials from Starch/Myristic Acid

cMS-NMs were fabricated as described in our previous studies with a few modifications [18,19]. We added starch and myristic acid (1:1 ratio) in 10 mL of deionized water to form a dispersion. Afterwards, the mixture was transferred to a non-stick pan. Then, the mixture was fried at a high temperature. As a result of frying, a dehydrated myristic acid/starch complex (dMS) was formed. The dMS was kept in a furnace at 250 °C for 20 min. The final product was dispersed in deionized water and chloroform. A liquid–liquid extraction technique was exploited to separate the immiscible portion. After the purification process, the obtained chloroform fraction (cMS-NMs) was used to examine their physicochemical and cytotoxic features.

### 2.3. Characterization of Starch—Myristic Acid Complex

FTIR spectroscopy was used to determine the surface functionalities of the cMS-NMs. Thermogravimetry analysis was conducted to investigate the cMS-NMs’ thermal degradation properties. We examined particle size, agglomeration, and zeta potential using a Zetasizer (Malvern, PA, USA). The particle size and morphology of the cMS-NMs were examined using a transmission electron microscope (JEOL 3010, Tokyo, Japan).

### 2.4. Cell Culture

The THP-1 cells were used as a human monocyte model to evaluate cytotoxicity. The cells were cultured in the RPMI supplemented with 10% FBS and 1% streptomycin/penicillin antibiotics. The cells were maintained under humidified conditions at 37 °C in a CO_2_ incubator (Thermo Scientific, Waltham, MA, USA).

### 2.5. Determination of Cytotoxic Properties

The cytotoxic features of the cMS-NMs were assessed in THP-1 cells using the MTT (3-(4,5-dimethylthiazol-2-yl)-2,5-diphenyltetrazolium bromide) assay [34]. Approximately 1.5 × 10^4^ cells were cultured in a 96-well culture plate by incubating at 37 °C for 24 h in a CO_2_ incubator. The THP-1 cells were treated with different concentrations of cMS-NMs (0, 50, 100, 150, 200, and 250 µg/mL) for 24 and 48 h. MTT dye (5 mg in 1 mL of PBS) was added to each well and incubated overnight. Then, plates were centrifuged and the media carefully discarded, and 200 µL of DMSO was added to each well to dissolve the purple-colored formazan crystals. Following that, a multi-well plate reader (Bio-Rad Laboratories, Hercules, CA, USA) was used to measure the absorbance of the plate at 570 nm (measurement) and 630 nm (reference). The obtained values was used to calculate the cell viability percentage using MS Excel 2010 (Microsoft Corporation, Redmond, WA, USA).

### 2.6. Microscopic Studies

The cMS-NMs’ impact on the THP-1 cells’ cellular and nuclear architectures was examined using bright-field and fluorescence microscopy. Cells were plated (≈4 × 10^4^ cells/well) in a 24-well plate and cultured in a CO_2_ incubator. Subsequently, the human monocyte cell line was exposed to different concentrations of cMS-NMs (0, 100, and 150 µg/mL). After 24 and 48 h treatment, the cell morphological alterations were observed using a bright-field microscope. For fluorescence microscopic examination, the media were carefully removed and the cells stained with an instant fluorescent AO/EB dye. Observations of cytoplasmic and nuclear morphology were performed using a fluorescent microscope fitted with dual filters (excitation: 488 nm; emission: 525 nm) (Carl Zeiss, Jena, Germany).

### 2.7. Gene Expression

To assess the impact of the cMS-NMs on gene expression (IL1B, MIF, BAK, and NFKB1A) in the THP-1 cell line, we used a SYBR Green/ROX master mix kit (Qiagen, Valencia, CA, USA) to performed the gene expression analysis. The analysis was conducted in real time using a qPCR system (Applied Biosystems 7500 Fast, Foster City, CA, USA). Glyceraldehyde-3-phosphate dehydrogenase (GAPDH) as the reference gene was utilized to normalize the expression levels of the target genes. The real-time qPCR reaction mixtures were prepared in a 96-well PCR plate containing 25 µL, which included 2.5 µL of assay primers (10×), 12.5 µL of master mix, and 10 µL of template cDNA (500 ng). The reaction cocktail was dispensed into an optical PCR plate and underwent the following steps for RT-qPCR: (1) initial denaturation at 95 °C for 5 min, (2) 95 °C for 30 s, and (3) 60 °C for 10 s for 40 cycles. CT analysis was employed to interpret the collected data, comparing cells exposed to the cMS-NMs with control samples. The mean fold changes over control were calculated using the following formula: ΔΔCt = ΔCt (treated) − ΔCt (untreated control). The outcomes were represented using 2^−ΔΔCt^.

### 2.8. Gene Marker Selection

To identify potential gene markers, we utilized the STRING database, a powerful bioinformatics resource that integrates and predicts protein–protein interactions. STRING provided valuable insights into the interactions between genes, enabling the exploration of intricate relationships among them.

### 2.9. Statistical Analysis

The collected data were analyzed using Microsoft Excel software, and the results, including the mean of three independent replicates and their standard deviations (Mean% ± SD%), were presented. The statistical significance (*p* value) of mRNA relative comparison over the control was determined through a Student’s *t*-test, which was conducted using Microsoft Excel software.

## 3. Results and Discussion

Lipids from animals and plants play a crucial role in the preparation of fast food, contributing to its flavor, texture, heat transfer, moisture retention, and energy content. There are a variety of fatty acids, including myristic acid, that are commonly found in various plant and animal fats. Myristic acid, a saturated fatty acid containing a 14-carbon straight chain, holds many roles in the food industry, serving as a multipurpose food additive and flavor adjuvant [35]. Alongside fats, starch—primarily sourced from potatoes, corns, rice and tapioca—which is mainly composed of amylose and amylopectin, is frequently used in various fast-food cooking methods such as thickening, coating, and binding. When exposed to high-temperature cooking techniques like frying, grilling, and deep frying, starch effectively interacts with fatty acids (i.e., myristic acid), generating a complex structure [18,19].

Advanced cooking methods can initiate a series of molecular changes that lead to the formation of nanomaterials/intermediate materials through the breakdown and/or complexation of myristic acid and starch. As a result of the heating process, carbonaceous fragments are released from myristic acid and starch molecules. Subsequently, these liberated fragments might undergo essential chemical alterations, transitioning through cyclization, aromatization, and polymerization [36,37,38,39]. As a result of these modifications, tiny carbon-rich clusters are formed, which eventually amalgamate and aggregate to form carbon-based nanoparticles on the surface of the foods [11,40,41,42,43]. It is critical to understand these molecular changes as well as the process of producing nanoparticles as a result of the pyrolysis and/or browning reaction of myristic acid and starch molecules. Some of the earlier studies reported these formation of these carbon-based nanomaterials and the consequences they exert on the physiology of the human system [24,44,45].

In this present study, the created dMS complex underwent dehydration and charring at 250 °C. The resultant myristic acid/starch-based nanomaterials (cMS-NMs) were isolated using a liquid–liquid extraction process. Figure 1 shows the FTIR spectra of the isolated cMS-NMs. Notably, this sample displays distinct absorption peaks, indicating specific features. Peaks at 3442 cm^−1^ and 2919 cm^−1^ were observed in the cMS-NMs, which can be attributed to OH groups and methylene groups linked with various groups of chemicals including HCA, PHA, and others. Notably, the asymmetric stretching vibrations of the methylene group and the carbonyl group of the myristic acid were observed as distinct absorption bands at 2852 and 1708 cm^−1^, respectively. Between 1410 and 1460 cm^−1^, distinct peaks were identified, signifying the occurrence of C-H bending vibration. These results clearly indicate the presence of myristic acid (MA) and starch (S) in the cMS-NM’s surface. Studying the surface chemistry of carbon nanomaterials poses significant challenges. These nanosized materials possess various functional groups that interact with cells in unique ways. For instance, carbon particles containing OH groups exhibit higher uptake by cells. Moreover, the presence of OH groups can contribute to the overall charge distribution on the particle surface. This alteration in surface charge may influence the electrostatic interactions between the carbon particles and the cell membrane, affecting the efficiency of cellular uptake mechanisms such as endocytosis [24,46]. Similarly, the interaction of cells with CH3 methyl groups differs from other functional groups. When methyl groups are added to the surface of the cMS-NMs, they can alter the physicochemical properties of the nanoparticles, influencing parameters such as hydrophobicity, charge, and reactivity. The hydrophobic nature may influence the uptake of nanoparticles by cells, potentially impacting the cellular internalization and subsequent intracellular processes [16,47].

The transmission electron microscopic images of the cMS-NMs are shown in Figure 2. The images revealed that the cMS-NMs possessed tiny spherical particles of 10–40 nm in size. The morphological features confirmed the presence of nanomaterials in the chloroform fraction. These nanomaterials are also formed during advanced cooking techniques. To examine the particle size distribution and zeta potential of the cMS-NMs, the dynamic light scattering (DLS) method was employed (Figure 3a,b). The average particle size of the cMS-NMs was determined to be 230 nm, indicating their nanoscale dimensions. Additionally, the zeta potential of the cMS-NMs was found to be −13 mV. Zeta potential is a measure of the electric charge present at the particle’s surface and provides information about the stability and potential interactions of nanoparticles in a solution. A negative zeta potential value, such as the −13 mV observed for the cMS-NMs, suggests the presence of negatively charged particles. This information is crucial for understanding the colloidal stability and behavior of cMS-NMs. Nevertheless, the results obtained from the DLS analysis were not in agreement with those obtained from the TEM analysis; agglomeration or aggregation may have contributed to this inconsistency. As a result of physicochemical processes, characteristics attributed to nanomaterials are formed during the high-temperature food process, including frying, roasting, and more. This study indicates that food products undergo a variety of modifications, both intended and unintended, as a result of modern cooking techniques. When food is exposed to high temperatures during cooking, complex chemical transformations occur at the molecular level. These transformations can include the denaturation of proteins, carbohydrates, and lipids. These processes can result in the formation of new molecular arrangements and interactions, leading to the generation of nanoscale structures [18,19].

In addition, the thermal behavior of the acquired cMS-NMs was assessed. Substantial weight loss was observed between 160 and 240 °C. Approximately 90% weight loss was observed in the cMS-NMs when subjected to a temperature of 300 °C, as depicted in Figure 4. This significant weight loss indicates the thermal decomposition or degradation of the cMS-NMs at high temperatures. The thermal gravimetric analysis provides valuable insights into the thermal stability and decomposition behavior of cMS-NMs. Based on the observed patterns of weight loss, we have concluded that the nanomaterials’ organic components contribute to the complex thermal processes.

Subsequently, we conducted investigations on the potential impact of the prepared cMS-NMs’ biological effects and potential risks, utilizing in vitro assays. The THP-1 cells were used as the human monocyte in vitro model. A cell viability assay was employed to study the cMS-NMs’ impact on the THP-1 cell viability and proliferation. The cell viability results are represented in Figure 5. After 250 µg/mL of cMS-NMs exposure to the THP-1 cells, we found cell death of approximately 95% at 24 h and 95% at 48 h. At lower-dose (100 µg/mL) treatment, above 50% of cell viability reduction was found. Notably, the cMS-NMs significantly inhibited the proliferation of the THP-1 cells. The impact on cell growth can be attributed to different factors associated with the special properties of cMS-NMs. The surface properties, size, and shape of these nanomaterials can have a significant role in determining how they specifically affect the growth of THP-1 cells.

The cell viability study revealed that the cMS-NMs decreased cell viability in a time- and dose-dependent manner. These results enhanced our interest in regulating the mode of cell death triggered by the cMS-NMs. Thus, we further investigated cytological changes after the cMS-NMs exposure to the THP-1 cells (Figure 6). In the bright-field microscopic images, we noticed cell shrinkage, disintegration, and swelling in the cMS-NMs-treated cells, but no defects were noticed in the control. These results clearly showed that the cMS-NMs triggered significant cytological changes in the THP-1 cells even at low (100 µg/mL) and moderate (150 µg/mL) doses. These morphological changes led to cellular damage (Figure 6). Alterations in the cell morphologies of the THP-1 cells is evidence of the cMS-NMs’ toxic effect. We used AO/EB dual dye to differentiate between apoptotic and necrotic cell death in the THP-1 cells. Upon staining the cells exposed to the cMS-NMs, we observed distinct morphological changes in their cytoplasms and nuclei. Specifically, we noticed nuclear and chromatic condensation, as well as significant DNA fragmentation in these cells. DNA fragmentation is a characteristic feature of apoptosis. A certain percentage of cells had nuclear condensation, which involves the shrinkage and compaction of the nucleus. These changes in cellular structure are indicative of apoptotic cell death. These outcomes of this study suggest that cMS-NMs induce cell death through both apoptosis and necrosis. It appears that the cytotoxicity of cMS-NMs is influenced by the unique nature of the nanomaterials and the cell phenotype. Carbon-based nanomaterials have the ability to interact with components inside cells, affecting important processes such as cellular signaling and gene expression, with special reference to negative immune response.

In Figure 7, three potential clusters are illustrated based on gene interaction networks. It is evident from this visual representation that the IL1B and NFKBIA genes are interconnected with a wide variety of other genes. This complex network of connections highlights the complex interactions and relationships between these key genes and their involvement in a wide variety of biological processes. Based on this, an impact of the cMS-NMs on the gene expression of the THP-1 was studied (Figure 8). Essential markers, such as the IL1B, MIF, BAK, and NFKB1A genes, are crucial components in the regulation of immune response, macrophage induction, and cell death, contributing to impaired cell processes. IL1B, also called interleukin-1 beta, is a special cytokine protein that helps control the immune system and responses to inflammation when triggered by external agents like cMS-NMs [48,49,50]. Upon the induction of cMS-NMs, THP-1 monocyte cells may alter the expression of IL1B significantly. There are a variety of functions associated with this cytokine; it acts by altering immune cell recruitment, mediating and stimulating the release of other inflammatory mediators. As a result of potentially harmful substances such as cMS-NMs, it plays an important role in initiating and amplifying the inflammatory cascade that may result in abnormal pathophysiology [51]. The macrophage migration inhibitory factor (MIF) is a cytokine that helps regulate the immune system and the process of inflammation in our bodies. The MIF plays an important role in the regulation of macrophages and different immune cells’ function by acting as a pro-inflammatory mediator [52,53]. Notably, the MIF helps different cells in the immune system communicate and coordinate their actions. Upon exposure to foreign or toxic materials like cMS-NMs, THP-1 monocytes may upregulate MIF expression significantly. The MIF can modulate many different cellular processes, including the pro-inflammatory stimuli, the migration of immune cells, and their activation [54]. BAK (BCL2 Antagonist/Killer 1) is a pro-apoptotic protein involved in the regulation of programed cell death (apoptosis) [55]. However, BAK did not alter its gene expression pattern in response to the cMS-NMs. Therefore, these cMS-NMs may not lead to the activation of apoptotic pathways through the BAK pathway. It should also be noted that NFKB1A, which is a subunit of the NF-kappa B transcription factor, plays a crucial role in regulating immune and inflammatory responses during inflammation [34,56,57]. When cells are exposed to foreign or toxic materials like cMS-NMs, NFKB1A can be modulated significantly. It plays a crucial role in regulating the expression of different genes that are involved in our immune system and responses to inflammation. Upon examination, it was noted that the cMS-NMs initiated an inflammatory response and led to changes in cellular and nuclear morphology in the THP-1 cells. Specifically, the cMS-NMs prompted necrotic cell death. Analysis of the gene expression patterns revealed alterations caused by the cMS-NMs. The study’s observations underscore the necessity for additional investigation to comprehend the potential health impacts of cMS-NMs. Overall, these genes play integral roles in the cellular response of THP-1 monocytes to cMS-NMs. A multitude of factors contribute to the process of initiating, regulating, and executing immune and inflammatory responses through production of different cytokines and the coordination of numerous cellular processes to counteract potential threats to cells and organs.

## 4. Conclusions

Upon exposure to high-temperature cooking methods, such as frying, grilling, and deep frying, at temperatures reaching 250 °C, a complex structure forms from the combination of starch and myristic acid. Analysis reveals that these cMS-NMs display spherical characteristics, with a diameter spanning from 3 to 60 nanometers, as identified through morphological analysis. cMS-NMs were found to induce an inflammatory response and provoke alterations in cellular and nuclear morphology in THP-1 cells. The cMS-NMs triggered both apoptotic and necrotic cell death. A study of gene expression patterns revealed that the cMS-NMs altered the IL1B and NFKB1A gene expression profiles. These study findings highlight the necessity for further research to understand the health implications of cMS-NMs in in vivo models. This study clearly demonstrates the formation of foreign or toxic nanomaterials resulting from high-temperature heating. Specifically, the research strongly indicates that cMS-NMs elicit an inflammatory response. According to these findings, it is essential to avoid such heat-induced foodborne particles, found in a wide variety of junk foods, as they can cause health problems. Additionally, food preparation methods must be reevaluated and safer alternatives prioritized to prevent potential health risks.

## Figures and Tables

**Figure 1 foods-13-00554-f001:**
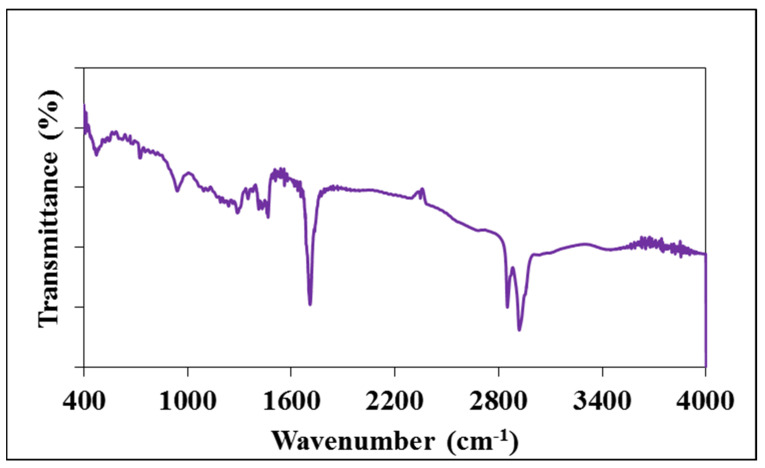
FTIR spectra of myristic acid/starch complex-derived nanomaterials.

**Figure 2 foods-13-00554-f002:**
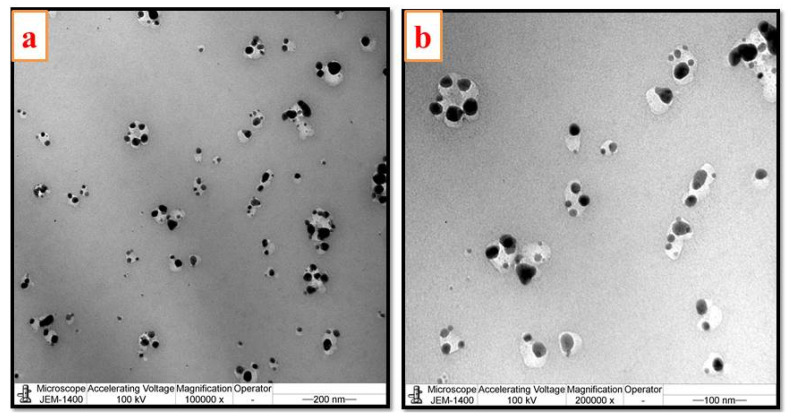
Transmission electron micrographs of myristic acid—potato starch complex-based nanostructured materials derived by (**a**,**b**) chloroform fraction.

**Figure 3 foods-13-00554-f003:**
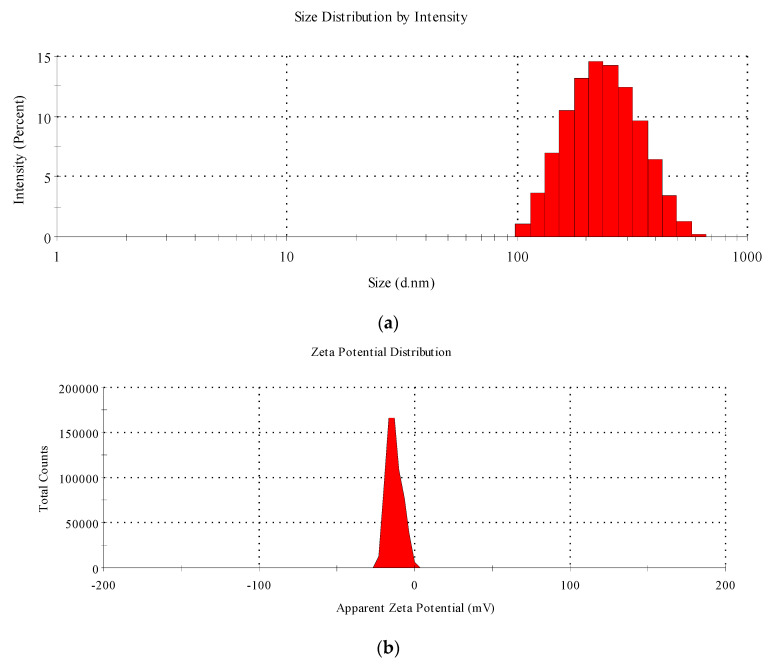
(**a**) Particle size distribution analysis of the cMS-NMs. (**b**) Zeta potential distribution analysis of the cMS-NMs.

**Figure 4 foods-13-00554-f004:**
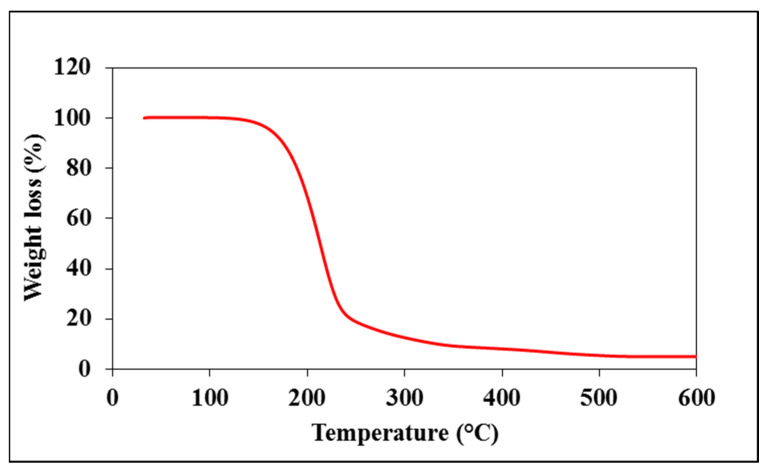
Red line depicted thermogravimetric curve of myristic acid/starch complex-derived nanomaterials.

**Figure 5 foods-13-00554-f005:**
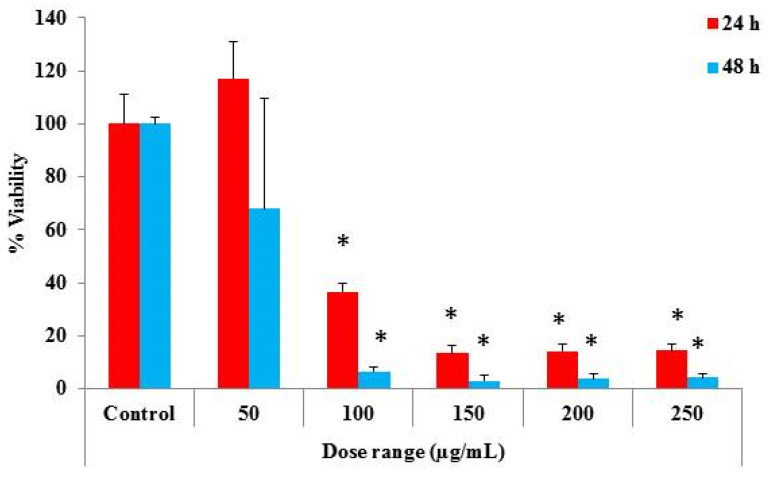
The effects of cMS-NMs derived from myristic acid/starch complexes on the viability of human monocyte cells (THP-1) over a period of 24 and 48 h. The data presented in the figure are the result of conducting the experiments in triplicate (* significance level *p* < 0.05).

**Figure 6 foods-13-00554-f006:**
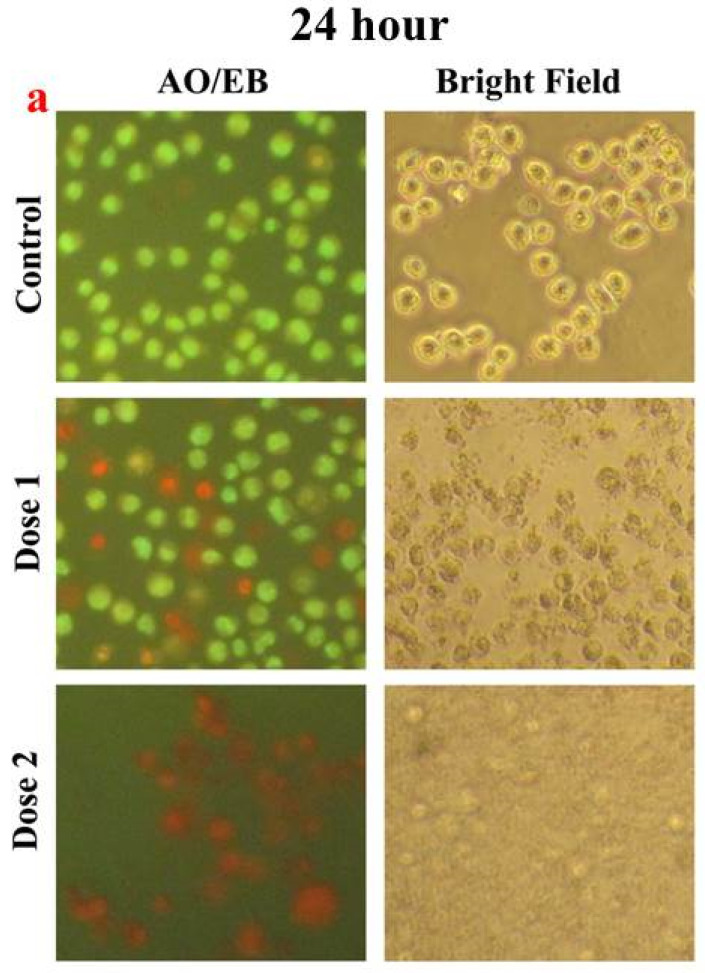
(**a**) Fluorescence and bright-field microscopic images of THP-1 cells after the exposure to cMS-NMs for 24 h. (**b**) Fluorescence and bright-field microscopic images of THP-1 cells after exposure to cMS-NMs for 48 h.

**Figure 7 foods-13-00554-f007:**
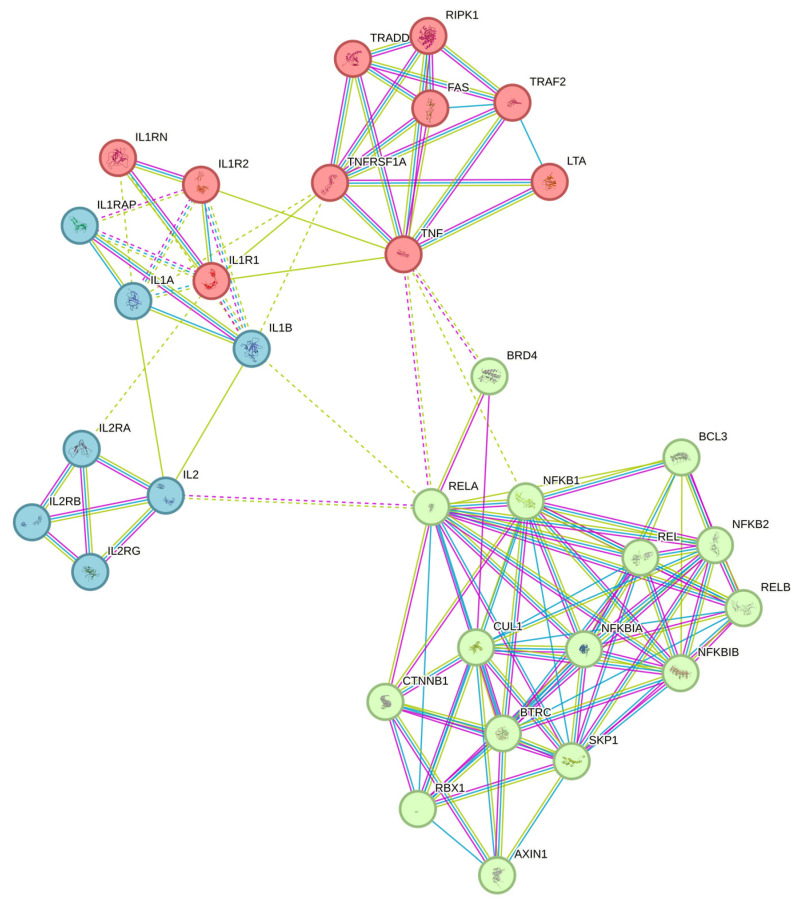
Visualization of gene network analysis (blue, green and red colored nodes indicate different clusters; green edges—gene neighborhood; red edges—gene fusions; blue edges—gene co-occurrence; pink edges—experimentally determined) performed using the STRING network analysis database.

**Figure 8 foods-13-00554-f008:**
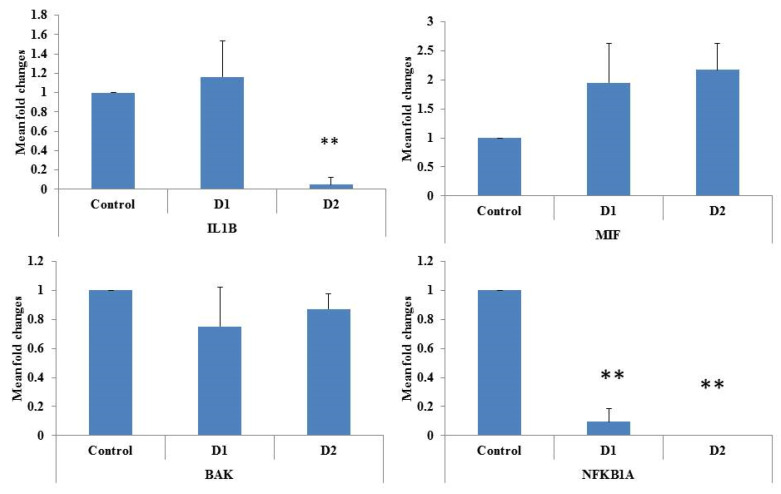
The mRNA expression level of THP-1 cells was analyzed after exposing them to cMS-NMs. Experiments were performed in triplicate. To analyze the gene expression level, an internal reference gene called GAPDH was used. (** significance level *p* < 0.05).

## Data Availability

Data are contained within the article.

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
