# Peer review of "Understanding the Interaction between Nanomaterials Originated from High-Temperature Processed Starch/Myristic Acid and Human Monocyte Cells"

_foods, 2024, doi:10.3390/foods13040554_

Round 1

Reviewer 1 Report

Comments and Suggestions for Authors

Periasamy et al report the synthesis of nanomaterials using starch and myristic acid (cMS-NMs) prepared at high temperature. The nanomaterials are purified by liquid-liquid extraction and their physicochemical property and cytotoxicity were evaluated.

The topic is of interest to the audience of Foods.

The reviewer has a few questions :

- the authors recover the cMS-NMs in the chloroform fraction. Can it be a problem for further use in biological experiments ?

- what is the yield of the synthesis ?

- are the cMS-NMs colloidaly stable in the biological medium used for the cells viability experiments ?

- are the cMS-NMs chemically stable in the cells or do they degrade?

- concerning the impact of NMs on gene expression, the authors should cite Lécuyer et al, Nanoscale 2022, 14, 15760-15771.

Author Response

Reviewer 1

Reviewer comment 1  :  The authors recover the cMS-NMs in the chloroform fraction. Can it be a problem for further use in biological experiments ?

Author’s response: Thank you for your insightful question regarding the recovery of cMS-NMs in the chloroform fraction. We want to clarify that we have thoroughly processed and evaporated the chloroform solvent, converting it into an emulsion to address its non-polar properties. We believe this modification mitigates any potential issues for the further use of cMS-NMs in biological experiments.

Reviewer comment 2 :  What is the yield of the synthesis ?

Author’s response: Thank you for your inquiry regarding the yield of the synthesis. We initiated the process with 5g of myristic acid and 5g of starch, resulting in a crude yield of 2.6g of CHCl3 fraction alone. After processing, the final yield was slightly lower.

Reviewer comment 3 :  Are the cMS-NMs colloidaly stable in the biological medium used for the cells viability experiments ?

Author’s response: Yes, the cMS-NMs are colloidally stable in the biological medium employed for the cell viability experiments. We achieved this stability by converting them into a nanoemulsion using non-toxic doses of bile salts, ensuring free from clumps or aggregates. This modification enhances the reliability of our experimental conditions.

Reviewer comment 4 : Are the cMS-NMs chemically stable in the cells or do they degrade?

Author’s response: Thank you for your insightful question regarding the chemical stability of cMS-NMs within cells. In our investigation, we focused on the basic interactions with human monocyte models, revealing that cMS-NMs uptake into cells induces cell death. While we have confirmed this initial observation, we acknowledge the importance of further studies to analyze the complete fate of cMS-NMs, and we are actively conducting ongoing research in this direction.

Reviewer comment 5 : Concerning the impact of NMs on gene expression, the authors should cite Lécuyer et al, Nanoscale 2022, 14, 15760-15771.

Author’s response: Thank you; we have included the reference to Lécuyer et al., Nanoscale 2022, 14, 15760-15771, in the appropriate section of our manuscript.

Reviewer 2 Report

Comments and Suggestions for Authors

This manuscript discusses the interaction between starch and myristic acid at 250 C, resulting in the formation of a nanoscale product. Extraction of the resulting nanoparticles in chloroform causes death on contact of THP-1 cells used as a human monocyte model. This topic, the approaches and methods used, and the results obtained are very close to those previously published by the authors: Morphological, Thermal, and Cytotoxicity Features Assessment of Starch–Myristic Acid Complex-Derived Nanostructured Materials Mushawah Abdullah Almushawah, Vaiyapuri Subbarayan Periasamy, Jegan Athinarayanan, Maha Alhussain, Abdulrahman Alwarthan, Ali A. Alshatwi / Starch. https://doi.org/10.1002/star.202200239 ; Fabrication of Myristic acid – Potato Starch Complex Nanostructures and Assessment of Their Cytotoxic Behavior Mushawah Abdullah Almushawah,  Athinarayanan, Vaiyapuri Subbarayan Periasamy, Ali Alshatwi / Journal of the Science of Food and Agriculture https://doi.org/10.1002/jsfa.13071.

The manuscript did not present the novelty of their perspectives to claim the difference from the existing articles. The authors' use of a different solvent or different cell cultures is, in my opinion, not sufficient to consider this work as original.

Furthermore, the product obtained by the authors (cMS-NMs complex) is not sufficiently characterised. The IR spectroscopy and thermal analysis data provided by the authors are not relevant. Additional other methods are required. The IR spectrum coincides with that of myristic acid and, according to the thermal analysis data, the complex decomposes completely at 250 C. But this is the temperature at which it was obtained. How can this be explained? What is the interaction between starch and myristic acid? What is the mechanism of this process?

The manuscript cannot be recommended for publication due to lack of novelty. The paper has serious flaws and additional experiments are needed.

Author Response

Reviewer 2

Reviewer comment 1 : This manuscript discusses the interaction between starch and myristic acid at 250 C, resulting in the formation of a nanoscale product. Extraction of the resulting nanoparticles in chloroform causes death on contact of THP-1 cells used as a human monocyte model. This topic, the approaches and methods used, and the results obtained are very close to those previously published by the authors: Morphological, Thermal, and Cytotoxicity Features Assessment of Starch–Myristic Acid Complex-Derived Nanostructured Materials Mushawah Abdullah Almushawah, Vaiyapuri Subbarayan Periasamy, Jegan Athinarayanan, Maha Alhussain, Abdulrahman Alwarthan, Ali A. Alshatwi / Starch. https://doi.org/10.1002/star.202200239 ; Fabrication of Myristic acid – Potato Starch Complex Nanostructures and Assessment of Their Cytotoxic Behavior Mushawah Abdullah Almushawah,  Athinarayanan, Vaiyapuri Subbarayan Periasamy, Ali Alshatwi / Journal of the Science of Food and Agriculture https://doi.org/10.1002/jsfa.13071. The manuscript did not present the novelty of their perspectives to claim the difference from the existing articles. The authors' use of a different solvent or different cell cultures is, in my opinion, not sufficient to consider this work as original.

Author’s response: Yes, we acknowledge the similarity with our previous studies involving starch and myristic acid complexes. In this manuscript, we have taken a distinct approach by interacting carbon nanomaterials with human immune cells, yielding diverse results, especially in cell phenotypic specificity. The extensive data generated in this study was deemed too substantial for inclusion with our prior works on human stem cells. Therefore, we intend to communicate these findings separately as a new manuscript. Additionally, we have initiated in-depth investigations into the uptake and fate of these carbon materials.

Reviewer comment 2 : Furthermore, the product obtained by the authors (cMS-NMs complex) is not sufficiently characterised. The IR spectroscopy and thermal analysis data provided by the authors are not relevant. Additional other methods are required. The IR spectrum coincides with that of myristic acid and, according to the thermal analysis data, the complex decomposes completely at 250 C. But this is the temperature at which it was obtained. How can this be explained? What is the interaction between starch and myristic acid? What is the mechanism of this process?

Author’s response: Thank you for your insightful remarks and considerations regarding the characterization of the cMS-NMs complex. We value your feedback on the IR spectroscopy and thermal analysis data and acknowledge the necessity of employing additional methods to improve our characterization efforts. Speciating carbon particles poses inherent challenges, and we are actively exploring alternative surface characterization techniques for inclusion in our forthcoming manuscript on the uptake and fate of the cMS-NMs complex. Regarding the decomposition temperature of 250°C, we fully recognize the need for a thorough explanation and are currently conducting further investigations to elucidate the interaction between starch and myristic acid, including the underlying mechanism of this process. Your constructive suggestions will undoubtedly enhance the refinement of our manuscript.

Reviewer comment 4 : The manuscript cannot be recommended for publication due to lack of novelty. The paper has serious flaws and additional experiments are needed.

Author’s response: Thank you for your thorough review and valuable feedback. We appreciate your concerns regarding the novelty of the manuscript and acknowledge the challenges associated with characterizing carbon nanomaterials, particularly in terms of physico-chemical speciation. We concur with your assessment and fully recognize the need for additional experiments to address the identified flaws. Your constructive suggestions will guide our future work, and we are committed to making the necessary improvements to meet the standards for publication. Thank you for your time and insightful comments.

Round 2

Reviewer 2 Report

Comments and Suggestions for Authors

In my opinion, the scientific novelty of the manuscript is still very limited. there is a huge literature devoted to the study of starch-fatty acid interactions. The authors have not added anything new to these studies or to their previous articles. However, I leave the decision to the scientific editor.

Author Response

I sincerely apologize. The starch-fatty acid-based carbon particles may possess unique characteristics. Within the extensive family of carbon particles and carbon dots, there exist numerous distinct species that can be generated under high-temperature conditions. These particles can be differentiated based on factors such as size, shape, polarity, surface chemistry, and chemical nature. We believe that these unique particles have the potential to induce immunomodulatory effects. Our paper introduces novel particles and explores their interaction with immune cells. Our research represents the first identification of this particular species of carbon particles derived from starch-myristic acid under high-temperature conditions. Thank you for your consideration.